# Gate controlled anomalous phase shift in Al/InAs Josephson junctions

William Mayer[1], Matthieu C. Dartiailh[1], Joseph Yuan [1], Kaushini S. Wickramasinghe[1], Enrico Rossi[2] & Javad Shabani[1]*

In a standard Josephson junction the current is zero when the phase difference between superconducting leads is zero. This condition is protected by parity and time-reversal symmetries. However, the combined presence of spin–orbit coupling and magnetic field breaks these symmetries and can lead to a finite supercurrent even when the phase difference is zero. This is the so called anomalous Josephson effect—the hallmark effect of superconducting spintronics—which can be characterized by the corresponding anomalous phase shift. Here we report the observation of a tunable anomalous Josephson effect in InAs/Al Josephson junctions measured via a superconducting quantum interference device. By gate controlling the density of InAs, we are able to tune the spin–orbit coupling in the Josephson junction. This gives us the ability to tune the anomalous phase, and opens new opportunities for superconducting spintronics, and new possibilities for realizing and characterizing topological superconductivity.

[1] Center for Quantum Phenomena, Department of Physics, New York University, New York, NY 10003, USA. [2] Department of Physics, William & Mary, Williamsburg, VA 23187, USA. *email: jshabani@nyu.edu

Superconductivity and magnetism have long been two of the main focuses of condensed matter physics. Interfacing materials with these two opposed types of electron order can lead to many new phenomena. Recently these systems have drawn renewed theoretical and experimental attention in the context of superconducting spintronics[1] and in the search for Majorana fermions[2–5]. Novel heterostructures can provide the ingredients that are typically needed: superconducting pairing, breaking of time-reversal symmetry, and strong spin–orbit coupling.

A basic property of superconducting systems is that we can introduce a relation between charge current and the superconductor's phase. In the canonical example of a Josephson junction (JJ), this is the current-phase relationship (CPR). Systems with nontrivial spin texture generally introduce a relationship between charge and spin. In the case of spin–orbit coupling this can manifest in many ways including the spin Hall effect and topological edge states[6].

A hybrid system, combining spin–orbit coupling and superconductivity, results in a much richer physics where phase, charge current, and spin are all interdependent. This gives rise to new phenomena such as an anomalous phase shift which is the hallmark effect of superconducting spintronics[1]. In a standard JJ, the CPR always satisfies the condition $I(\phi = 0) = 0$, where $\phi$ is the phase difference between the two superconductors. This condition is protected by parity and time-reversal symmetries. However the presence of spin–orbit coupling along with the application of an in-plane magnetic field can break these symmetries[7]. This allows an anomalous phase ($\phi_0$), which means that with no current flowing there can be a non-zero phase across the junction or, conversely, at zero phase a current can flow[8,9]. This is also understood in the context of the spin-galvanic effect, also known as the inverse Edelstein effect. It states that in a normal system with Rashba spin–orbit coupling, a steady state spin gradient can generate a charge current[8]. When superconductivity is introduced, gauge invariance no longer prohibits a finite static current-spin response[9]. Consequently in the superconducting state, a static Zeeman field can induce a supercurrent, which can be measured as $\phi_0$[10,11].

Anomalous phase junctions were demonstrated in InSb nanowires in a quantum dot geometry[10] and more recently in JJ using $Bi_2Se_3$[11]. In the quantum dot realization the phase shift is gate tunable but is geometrically constrained and only supports a few modes and consequently small critical currents. In $Bi_2Se_3$, a topological insulator, large planar $\phi_0$-junction are possible, however $Bi_2Se_3$ is not gate tunable.

Our work is based on heterostructures formed by InAs and epitaxial superconducting Al[12] which have emerged as promising heterostructures not only for mesoscopic superconductivity[13] but also for the realization of topological superconductivity[14–16] and Majorana fermions[17]. This is due to the fact that the induced superconducting gap, $\Delta_{ind}$, in InAs can be as large as the one in Al[18], and InAs has large g-factor and spin–orbit coupling. As a consequence, JJ fabricated on this platform can have large critical current and high transparency[19,20]. Furthermore, one can control the strength of the spin–orbit coupling by tuning the density in the InAs via external gates[21].

## Results

### Device characterization.
Figure 1a shows a transmission electron microscope image of the heterostructure with false colors. We fabricate superconducting quantum interference devices (SQUID) consisting of two Al/InAs JJ's in parallel. The fabrication details were previously reported[18] and are detailed in "Methods". Figure 1b shows a tilted view scanning electron microscope image of

a device with false colors, and the device schematic is depicted in Fig. 1c. Both junctions are 4 μm wide (W) and 100 nm long (L) while the size of the SQUID loop is 25 μm². The high aspect ratio of the junction (W/L) yields devices that have many transverse modes and consequently large critical currents. Typical mean free path ($l_e$) in the semiconductor region is near $l_e \simeq 200$ nm and the superconducting coherence length ($\xi$) is estimated to be $\xi = 770$ nm[20]. The two junctions show small variations in normal resistance ($R_n$), $R_n^1 = 102\ \Omega$, $R_n^2 = 110\ \Omega$ and critical current ($I_c$) $I_c^1 = 4.4\ \mu A$, $I_c^2 = 3.6\ \mu A$ when gates are not activated. Gate voltage ($V_g$) varies the density of the InAs region thereby changing $R_n$ and $I_c$ of each JJ.

At low $V_g^2$ voltages, we can fully deplete JJ2 and turn our device from a SQUID to a single junction. This is confirmed by phase bias measurements performed by applying perpendicular magnetic field ($B_z$), shown in Fig. 1d, e. In Fig. 1d, when both junctions are at $V_g^1 = V_g^2 = 0$ V, we see characteristic SQUID oscillations with application of $B_z$. Superimposed on top of the fast SQUID oscillations is the much slower Fraunhofer diffraction pattern from each individual JJ. Conversely when $V_g^2 = -7$ V, in Fig. 1e, we observe only the Fraunhofer pattern indicating the presence of only a single JJ. This allows us to effectively study each JJ individually.

Individual JJs are characterized in in-plane magnetic field as shown in Fig. S1. We find $B_c = 1.45$ T for thin film Al in both junctions and is independent of the in-plane field direction. However, $I_c$ of both JJs show a strong asymmetry in in-plane magnetic field. We observe a stronger decrease in $I_c$ as a function of $B_x$ (field applied along the current direction). This is consistent with previous measurements on InAs 2DEG based JJ[17], and recent work suggests this could be related to the nature of spin–orbit coupling in the system[22]. Measurements of Fraunhofer pattern with increasing in-plane field show increasing asymmetry. Unlike previous studies this asymmetry is found to be independent of in-plane magnetic field direction. In addition, despite these distortions, the Fraunhofer pattern appears to remain periodic. Significant changes in current distribution, such as edge conduction, should alter the periodicity, specifically with respect to the central Fraunhofer peak. The absence of any such effect indicates a homogeneous current distribution at all fields. Figures and further discussion are presented in Supplementary.

### Current-phase relation and transparency.
Measurements of robust Fraunhofer pattern up to $B_y = 400$ mT are made possible in this system due to the large induced gap in the semiconductor region[19]. Using the product $I_cR_n/\Delta$, where $\Delta = 230\ \mu eV$ is the superconducting gap of the Al, the quality of the junction can be characterized. For the junctions used in this study we measure $I_c^1R_n^1/\Delta = 2$ and $I_c^2R_n^2/\Delta = 1.78$. Studies of CPR can also aid junction characterization, as a nonsinusoidal CPR indicates a highly transparent JJ. Measurements of skewed CPR have been demonstrated in InAs nanowires JJ[9] bismuth nanowires[23] and graphene devices[24]. The generalized CPR can be described by Eq. (1), where $\phi_t$ is the total phase across the junction, $\tau$ is the junctions transparency and we neglect any temperature dependence since all measurements are performed at $T = 20$ mK:

$$I(\phi_t) = I_c \frac{\sin\phi_t}{\sqrt{1 - \tau\sin^2\phi_t/2}}. \tag{1}$$

It should be noted that this expression describes the transparency of a single ballistic channel. However in our devices there are many conduction channels present (~300) so the transparency we extract should be considered an average over all the channels. In the absence of disorder each channel can be

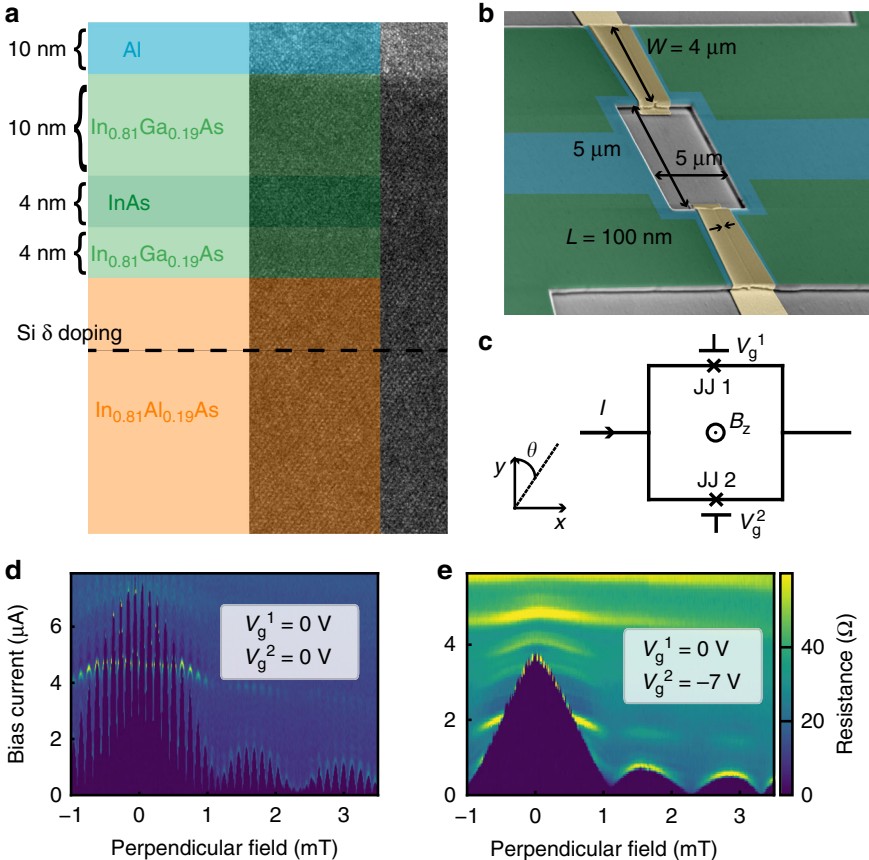

**Fig. 1 SQUID consisting of two gate-tunable InAs junctions. a** Sample stack description superimposed on large scale TEM image. **b** Colorized SEM image of a device similar to the one presented. The SQUID loop is about $5 \times 5\,\mu m$, both junctions have a gap of about 100 nm and are $4\,\mu m$ wide. **c** Schematic of the device. Each junction can be gated independently. The $x$ direction is defined in the plane of the sample along the current direction. **d** Resistance of the SQUID as a function of the perpendicular field and bias current with both gates set at 0 V. Typical fast SQUID oscillation of the critical current can be seen on top of the larger scale Fraunhofer pattern of the junctions. **e** Resistance of the SQUID as a function of the perpendicular field and bias current with $V_g^1$ set at 0 V and $V_g^2$ set at $-7$ V. The SQUID oscillations visible in **d** are completely absent and only the single junction Fraunhofer pattern is visible.

considered individually. If we assume a Gaussian distribution of transparencies we approximately recover the single channel result for the mean transparency. In a more realistic system finite disorder can mix channels and substantially alter the junction properties as will be discussed in the context of an anomalous phase shift below. To measure the CPR, we apply gate voltages to the junctions to create a highly asymmetric current configuration ($I_c^1 \approx 4I_c^2$). This effectively fixes the phase of the high current junction so we measure only the CPR of the lower current junction. Figure 2a shows resistance maps at $B_y = 50$ mT, $B_y = 200$ mT, and $B_y = 350$ mT in the CPR regime. At $B_y = 50$ mT the plot shows a forward skew indicating high JJ transparency. To fit the SQUID oscillations, we sum the contributions of each JJ with a phase difference due to applied $B_z$ and maximize the current with respect to the sum of the phases. The resulting fits are shown in Fig. 2a as orange overlays. The transparencies obtained from the fits are indicated in each plot. Measurement at $B_y = 350$ mT reveals the oscillations are more sinusoidal, indicating reduced transparency. The dependence of transparency on $B_y$ for JJ2 is shown in Fig. 2b. We observe near unity transparency at low fields, with a rapid decline above 200 mT. Both junctions show similar dependence of transparency on $B_y$. The mechanism leading to the decreased transparency as a function of $B_y$ is not well understood. Note that these fits are based on the assumption that the JJ CPR is captured by Eq. (1).

**Anomalous phase shift**. If we consider a single JJ with an anomalous phase, a typical current-biased measurement will show no measurable signature. When a JJ is current biased, the CPR dictates that the phase will change so the critical current is maximized. This means that any phase shift applied to such a system will be invisible once the current is maximized. A simple alternative which has been employed in previous studies of $\phi_0$ is to use a SQUID geometry, whose primary property is phase sensitivity. Even in a SQUID, any single scan generally has an phase offset obscuring the effect of $\phi_0$. In order to experimentally measure $\phi_0$, a phase reference is necessary. To this end we compare scans taken consecutively at the same field but changing $V_g$ of one JJ. The gate voltage varies both the density and strength of spin–orbit coupling which should change $\phi_0$. Figure 3 shows resistance maps taken at different $B_y$ for three $V_g^2$. By finding the phase shift between these different gate voltages we can measure the variation of $\phi_0$. This shift is most easily seen by comparing the positions of SQUID oscillation maxima at different $V_g^2$. To extract the phase difference we fit the data using a similar procedure as applied to the CPR of Fig. 2. The only adjustment is that we include $\phi_t = \phi + \phi_0$ in each CPR relation. In the case of a varying transparency, one could observe an apparent phase shift unrelated to $\phi_0$. However this shift would have the opposite sign on the positive and negative bias branches of the measurement. The data presented in Fig. 3 are symmetric in bias, which allows us to definitively separate the effects of transparency and a $\phi_0$ shift. A

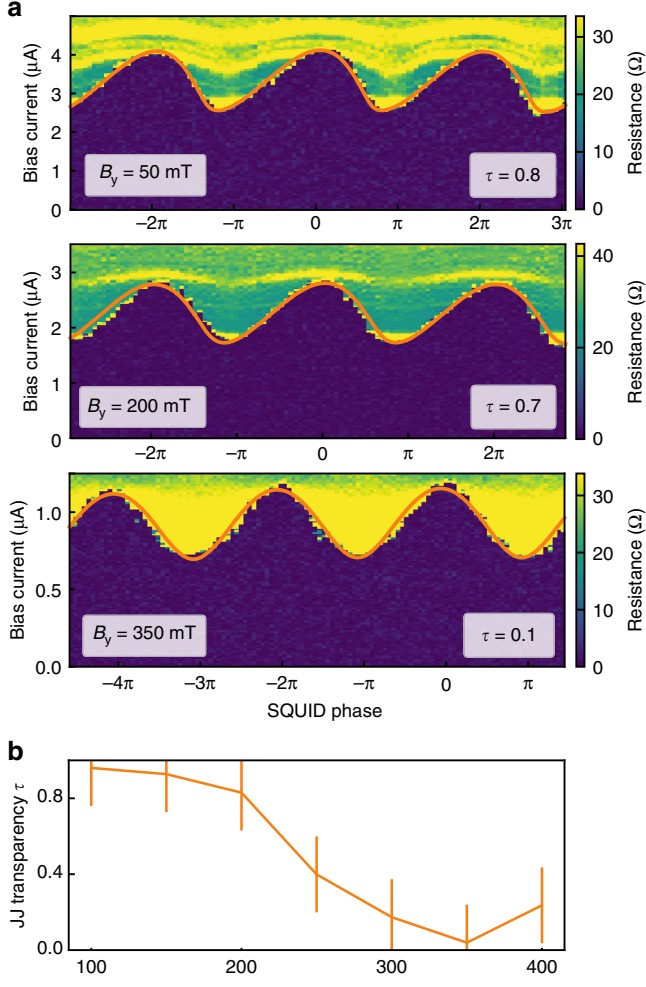

**Fig. 2 Current-phase measurement of SQUID. a** Resistance of the device as a function of the phase bias applied on the SQUID and the bias current in the presence of an in-plane field along the $y$ direction at $B_y = 50$ mT, $B_y = 200$ mT and $B_y = 350$ mT. $V_g^1$ is set to $-2$ V and $V_g^2$ to $-4.5$ V resulting in $I_c^1 \approx 4I_c^2$. The oscillation of the critical current present a visible forward tilt at 50 and 200 mT absent at 350 mT. **b** Evolution of the transparency of JJ2 as a function of the in-plane field $B_y$ as determined from fitting the SQUID oscillation at different gate and fields (see "Methods"). The error bars correspond to typical variations observed across different similar data sets.

more detailed description of the fitting can be found in "Methods".

The anomalous phase $\phi_0$ is expected to grow with the strength of the spin–orbit coupling. Previous work on InAs indicates that the Rashba spin–orbit coupling can be tuned from close to zero to as high as 180 meV Å, with apparent saturation at high densities[21]. This indicates that $\phi_0$ should be smallest at the lowest gate voltages. Consequently, we take $V_g = -4$ V as the reference scan which allows us to minimize the reference contribution to $\Delta\phi_0$, i.e., the difference $\phi_0(V_g) - \phi_0(-4V)$. Figure 4a shows how $\Delta\phi_0$, extracted from the fits, increases with gate voltage and saturates at higher $V_g$. In[21] it was shown that $\alpha$ increases as density ($n$) increases but that for low densities the relationship is nonlinear. This could explain the general $V_g$ dependence of $\phi_0$ since at low $V_g \alpha$ is increasing faster than $n$ leading to a rapid increase of $\phi_0$ versus $V_g$, while at higher $V_g$ the effect of $\alpha$ and $n$ cancel out and the $\phi_0$ dependence on $V_g$ weakens.

Several theoretical works have studied the interplay of spin–orbit coupling and time-reversal breaking fields in JJs. They provided scalings of $\phi_0$ with respect to material and geometry parameters[25–28]. Almost all the available theoretical works consider the long junction limit in which the distance $L$ between the superconductors is much larger than the coherence length $\xi$. In this limit, for a single transverse mode, theory predicts $\phi_0 = 4\alpha L\, E_z/(\hbar v_F)^2$ in the ballistic regime[26], and $\phi_0 = m^{*2}(\alpha L)^3\, E_z/(\hbar^3 v_F)^2$ in the diffusive regime[28], where $m^*$ is the effective mass and $v_F$ is the Fermi velocity.

Both analytic expressions reflect the fact that the anomalous Josephson effect is expected to be stronger as the ratio $L/\xi$ increases. However, by substituting in these expressions our material parameters, we find that both results return values of $\phi_0$ that are much smaller than what we observe. This is not surprising considering that in our devices $\xi \sim 770$ nm. In addition, both expressions are obtained in the limit of weak proximitized superconductivity, obtained by imposing a finite contact resistance at the interface. In addition, theoretical work in the short junction limit is generally restricted to nanowire systems with only a few conduction channels[29,30]. This leads to a geometry that is still drastically opposed to the current situation where $W \gg Ł$, which cannot be achieved in nanowires.

To understand the large value of $\phi_0$ in our devices it is important to first understand the affect of having a very large number of transverse modes. For a few of these modes $v_F$ is very small and therefore $L/\xi > 1$. Consequently these few modes can be described in a long diffusive limit, greatly increasing their contribution to $\phi_0$. Coupled with the fact that the proximity effect is strong in this system, this provides a qualitative explanation for the larger than expected values of $\phi_0$.

Figure 4b shows the dependence of $\Delta\phi_0$ on $B_y$ at a range of gate voltages. The strong agreement with linear fits confirms that $\Delta\phi_0$ is proportional to the Zeeman energy in agreement with theory[9]. With a more complete theoretical understanding in the limit of strong proximity effect, it should be possible to estimate the strength of spin–orbit coupling from the slope of the anomalous phase dependence. At the largest $B_y$ and $V_g$ measured we observe $\Delta\phi_0 > \pi/2$ setting a lower bound on $\phi_0$. It is possible to optimize both $L$ and $W$ of each JJ to increase $\Delta\phi_0$, and consequently $\phi_0$.

## Discussion

In summary, we have shown the capability to tune the anomalous phase shift of JJs formed by InAs and Al. This tunability results from the ability to vary the strength of the spin–orbit coupling via an external gate. The observation of a finite $\phi_0$ indicates a coupling of the superconductors phase, charge current, and spin in these heterostructures. We find $\phi_0$ to be proportional to the Zeeman energy, as expected, and its magnitude to be much larger than the currently available theoretical scalings. This is most likely due to the presence of a large number of conductions channels and the strong proximity effect in our system.

The capability to realize a large value of $\phi_0$ and to tune it is of great importance for applications in superconducting spintronics where large spin gradients can be used to realize phase batteries[1], and opens the possibility to generate, in a controllable way, spin gradients through Josephson currents or a phase bias. In addition, the observation that a significant $\phi_0$ can be present in InAs/Al heterostructures, and the fact that it strongly depends on the density of InAs, are directly relevant to efforts to realize topological superconducting states. In particular, the knowledge that an intrinsic phase difference $\phi_0$ can be present in InAs/Al JJs is of great importance for recent proposals to realize topological superconductivity in phase-controlled JJs[15,16].

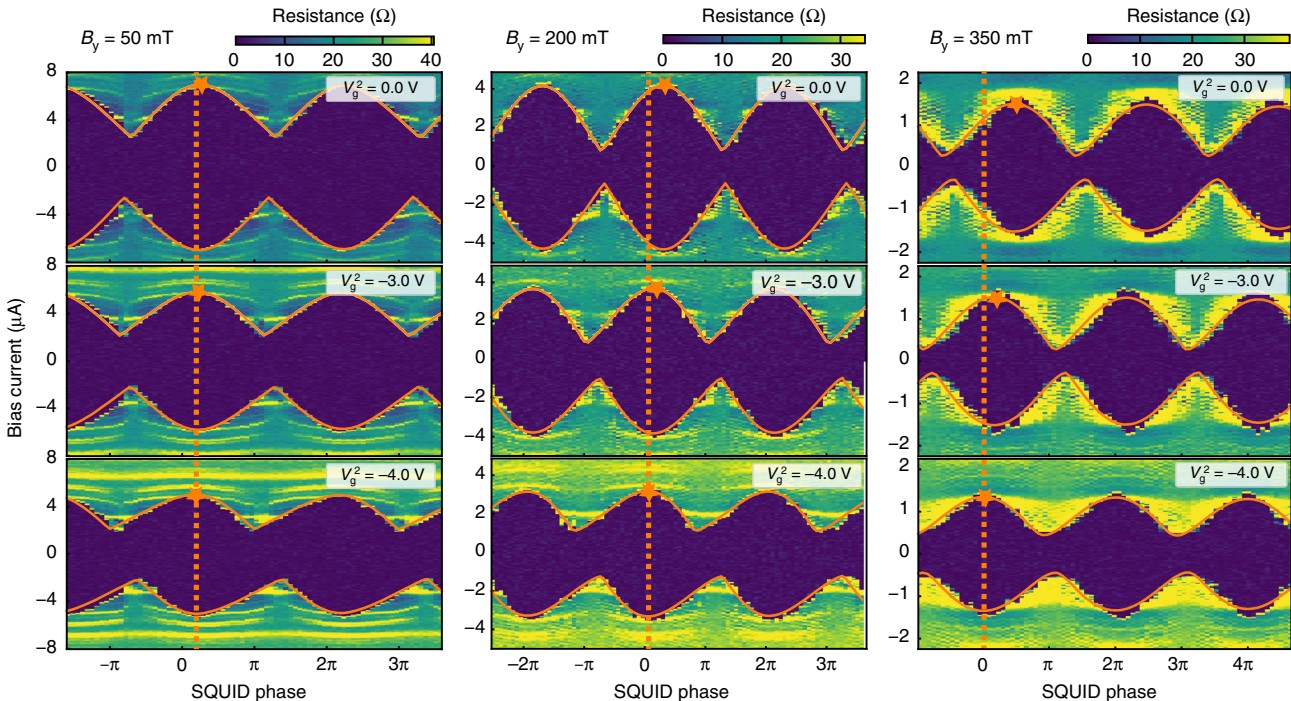

**Fig. 3 Resistance of the device as a function of the phase bias applied on the SQUID and the bias current at three different values of the in-plane field $B_y$ and three different values of $V_g^2$.** In all scans $V_g^1$ is set to $-2$ V. The dashed orange line indicates the position of the maximum of the oscillation at $V_g^2 = -4$ V. Orange stars indicate the position of the maximum at each field.

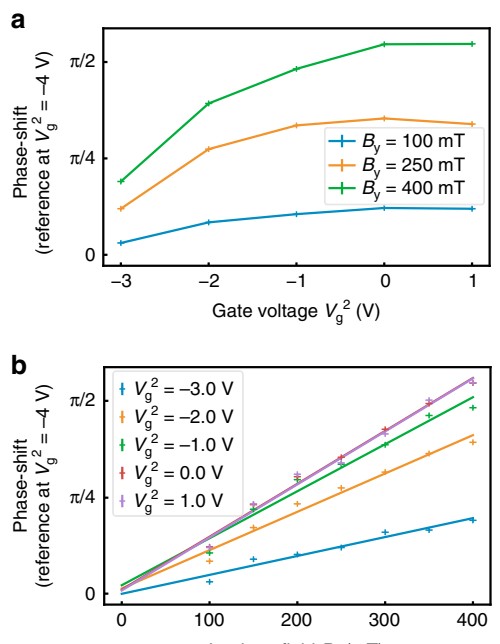

**Fig. 4 Tuning anomalous phase shift using gate voltage and in-plane magnetic field.** Evolution of the phase shift in JJ2 as a function of the gate voltage (**a**) and of the applied in-plane field along y (**b**). The phase shift $\Delta\phi_0$ is measured between the oscillations at a given value of $V_g^2$ and the ones at $-4$ V used as reference. In **b** the solid lines corresponds to linear fits to the measured phase shifts.

## Methods

**Growth and fabrication.** The structure is grown on semi-insulating InP (100) substrate. This is followed by a graded buffer layer. The quantum well consists of a 4 nm layer of InAs grown on a 4 nm layer of $In_{0.81}Ga_{0.25}As$ and finally a 10 nm

$In_{0.81}Ga_{0.25}As$ layer on the InAs which has been found to produce an optimal interface while maintaining high 2DEG mobility[21]. This is followed by in situ growth of epitaxial Al (111). Molecular beam epitaxy allows growth of thin films of Al where the in-plane critical field can exceed $\sim$2T[12].

Devices are patterned by electron beam lithography using PMMA resist. Transene type D is used for wet etching of Al and a III–V wet etch ($H_2O:C_6H_8O_7$: $H_3PO_4:H_2O_2$) is used to define deep semiconductor mesas. We deposit 50 nm of $Al_2O_3$ using atomic layer deposition to isolate gate electrodes. Top gate electrodes consisting of 5 nm Ti and 70 nm Au are deposited by electron beam deposition.

**Measurements.** All measurements are performed in an Oxford dilution refrigerator with a base temperature of 7 mK. The system is equipped with a 6:3:1.5 T vector magnet. All transport measurements are performed using standard dc and lock-in techniques at low frequencies and excitation current $I_{ac} = 10$ nA. Measurements are taken in a current-biased configuration by measuring $R = dV/dI$ with $I_{ac}$, while sweeping $I_{dc}$. This allows us to find the critical current at which the junction or SQUID switches from the superconducting to resistive state. It should be noted we directly measure the switching current, which can be lower than the critical current due to effects of noise. For the purposes of this study we assume they are equivalent.

**Fitting procedure.** As illustrated in Fig. 2, the junctions forming the SQUID display a saw-tooth like CPR characteristic of junctions with high transparencies, and this even at low gate. We hence model the CPR using Eq. 1 in which we neglect the temperature dependence which would only induce minor corrections. To model the SQUID pattern, we sum the contributions of two JJs with a phase difference and maximize (minimize for negative bias current) the current with respect to the sum of the phases. This requires the use of six parameters: the out-of-plane magnetic field to phase conversion factor, the transparency of each junction, the critical current of each junction (defined as independent of the transparency) and a phase. This represents a large number of parameters for fitting a single trace. To improve the accuracy of our procedure we consider multiple traces and reduce the number of parameters based on physical arguments.

Since we cannot experimentally access a reliable phase reference, we always compare measurements taken within a single magnetic field sweep, for different values of the gate voltage applied to one of the junction (referred to as the active junction). The second junction (idler) stays at a constant gate voltage. We can hence fix the amplitude of the idler current for a given parallel field.

Changes in the transparency of a junction can cause an apparent phase shift when considering only the positive bias current branch of the SQUID oscillation. However this apparent shift would have the opposite sign for the negative bias current branch. We have checked, as illustrated in Fig. 3, that the phase shift we observe is present with the same sign on both branches. As a consequence we can reasonably assume that the transparency of the junctions is constant over the gate

voltage range considered. This assumption allows us to use one transparency value per junction at a given field. The transparency value is better constrained in a CPR-like measurement and this is why, to have a well constrained problem, we combine data sets taken in both configurations: JJ1 as active junction and JJ2 as idler and JJ2 as the active junction and JJ1 as idler.

Considering measurements at $N$ parallel fields with $M$ different gate values in both configuration (JJ1 active/JJ2 active), we fit for each junction N transparencies, N amplitudes as idler, N × M amplitudes as active. Furthermore we extract $2 \times N \times M$ phases. Because the field to phase conversion factor depends only on geometrical considerations we use a single value for each configuration (We observed that for data sets taken several weeks apart we could see small changes in the field to phase conversion factor, that we attribute to the magnet. As a consequence we use different factors for data taken when tuning JJ1 or JJ2). For the most extensive dataset, presented in Fig. 4, $N = 7$ and $M = 6$. Similarly, we can also take into account the Fraunhofer envelope of the oscillation using two global parameters: a period and a phase.

By comparing the transparencies from independent measurements of JJ1 and JJ2 at a given magnetic field, we find that the junction transparencies are very similar. Hence, the data for Figs. 2a and 3 have been fitted using the equal transparencies assumption. The data for Figs. 2b and 4 have been fitted using the full method presented above but we focused on JJ2 results.

## Data availability

All data are available from the corresponding author upon reasonable request.

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

## Acknowledgements

This work is supported by DARPA Topological Excitations in Electronics (TEE) program and NSF. We acknowledge fruitful discussions with Igor Zutic and Alex Matos-Abiague. ER acknowledges support from ARO Grant No. W911NF-18-1-0290, NSF Grant No. DMR-1455233 CAREER, and ONR, and helpful discussions with Joseph Cuozzo and Stuart Thomas. This work was performed in part at the Advanced Science Research Center NanoFabrication Facility of the Graduate Center at the City University of New York.

## Author contributions

Samples were grown by K.W. and J.Y., and device fabrication was performed by W.M. and J.Y. Measurements and data analysis were performed by W.M. and M.D. Conception and data interpretation was done by W.M., M.D., E.R. and J.S. All authors were involved in writing and editing of the paper.

## Competing interests

The authors declare no competing interests
