## [Peer Review File · Nature Communications]

Reviewers' Comments:

Reviewer #1:

Remarks to the Author:

This paper reports on the fabrication of a SQUID based on two parallel hybrid Josephson junctions made from superconducting Aluminum electrodes deposited on an InAs quantum well. Two gates are used to control independently the carrier concentration and Rashba coefficient in the two Josephson junctions. Changing the gate voltage on one of the junction, they observe a phase-shift of the current-phase relationship when an in-plane magnetic field is present. This phase shift is attributed to the anomalous phase shift expected theoretically in presence of a finite Rashba coefficient and finite spin-polarization (here induced by the in-plane magnetic field). The data and their interpretation are convincing.

The anomalous phase shift has been observed recently in two other systems, InSb nanowires [Ref 4] and Bi₂Se₃ [Ref 5]. However, as described in their introduction, this observation of the anomalous phase shift in 2D hybrid Josephson junction with InAs quantum wells is important and timely given the major role of the InAs platform for Majorana physics.

Furthermore, this result on InAs, together with the recent result on Bi₂Se₃ [5], may motivate the search for the anomalous phase shift with many other materials presenting strong spin-orbit coupling.

Because of this, I believe that this paper should be of interest to a large community and I recommend its publication in Nature Comm.

I have some minor suggestions that the author may consider:

Main text: Page 2, col 2, beginning 2 paragraph :

Individual JJs are characterized in in-plane magnetic field (Add ref to Supplementary Fig. S1).

Main text: Page 2, col 2, beginning third paragraph :

where Δ is the superconducting gap of the Al. (Precise which numerical value you took for Δ)

Method section A. Growth and fabrication:

We deposit 50 nm of AlO₂ using atomic layer deposition to isolate gate electrodes. AlO₂ is not common, are you sure you didn't mean Al₂O₃ ? please check.

Reviewer #2:

Remarks to the Author:

The paper "Gate Controlled Anomalous Phase Shift in Al/InAs Josephson Junctions" is an experimental paper reporting the observation of a tunable anomalous Josephson effect measured using a SQUID device.

The anomalous Josephson effect refers to the possibility of having a finite Josephson current at zero phase difference between the superconducting leads, or the opposite, i.e. a finite phase difference at zero flowing current.

There are very few experimental evidences of this phenomenon. Moreover, in addition with respect to previous papers, in the present manuscript the authors show a tunability of the effect using a voltage gate.

I think that the paper deserves publication. However I have some points that I would like to be clarified prior to publication.

1) The authors compare with theoretical models explicitly addressing long junctions. It would be better to compare and cite the papers predicting the effect in the short junction limit: e.g. , J. Phys. Soc. Jpn. 82 (2013), Journal of Physics: Condensed Matter 27, 205301 (2015), Physical Review B 98, 144510 (2018).

2) In Fig. 2 the authors fit the I vs ϕ relation using Eq. 1, and conclude that the junction transparency is strongly affected by the external magnetic field. However they admit that the mechanism behind this dependence is not well understood.

I am not sure that their approach is flawless here. Usually a correct approach to estimate the transmission of the Junction is performing excess current measurements (and using the very well known Blonder Tinkham Klapwijk theory). To me it looks quite unusual that a SNS system can have 0.1 transparency (that value sometimes can be found even in SIS junctions!). The point here is that the expression in Eq. 1 is well suited in the case of a single channel SNS junction in the ballistic limit. If I understand correctly, the junctions studied have a very large number of channels, each of them can have a different transparency, and extrapolating the junction transparency using eq. 1 can be a bit oversimplified.

3) I agree with the authors that they observe a large effect (large ϕ_0). However their explanation in terms of the number of channels is only qualitative and not satisfactory. I suggest the authors to invest more efforts to give a microscopic description of their sample. They could be able to do that as there are theorists in the team!

Minor points:

1) The references are not sorted.

2) Caption of Fig. 2 The authors refer to panel (c). I guess it should be panel (b)

Reviewer #3:

Remarks to the Author:

The work by Mayer et al, reports measurements of Josephson junctions using InAs QW's coupled to epitaxial Aluminum. The work looks to establish what has been predicted theoretically in the literature that a phase shift in the current phase relation should develop due to spin-orbit interaction. The work uses a SQUID geometry to determine the phase. The observed phase shift appears to be considerably larger than what theory predicts in the limit of junction length larger than the coherence length. The authors speculate that their work is in the opposite limit which is outside the range of validity of existing theory.

While the experiments are carefully performed, I feel there is more evidence needed in order to ensure the observed phase shifts are indeed a result of spin-orbit interaction and not some other less interesting effect. Below I provide some of my concerns and comments:

1. Effects of parallel field on Fraunhofer patterns - In page 1 the authors enter a discussion on the behavior of the system as a function of B_x and write:

"Measurements of Fraunhofer pattern with increasing in-plane field show increasing asymmetry, but unlike previous studies this asymmetry is found to be independent of in-plane field direction. Additionally, despite the distortions, the Fraunhofer pattern appears to remain periodic, indicating a homogeneous current distribution at all fields. Figures and further discussion are presented in Supplemental."

I could not follow this statement and was unable to find the discussion in the supplemental material on current distribution.

The effects of B_x are complex due to what is known as the Doppler effect. The fact the SC is above the 2DEG leads to a unique winding of the phase along the junction, when an in-plane field is present and applied perpendicular to the edge of the SC. This could modify the Fraunhofer pattern substantially. While this Doppler effect should be strongest for B applied along X (perpendicular to the edge of the SC) it could also have an increasing effect when the field is applied along Y . This comes in as random phases generated along the junction due to lithographic roughness of the superconducting edges. The effect along X is known to strongly modify the current flow pattern in particular near the edges of the junction.

I am therefore wondering what some of the consequences of this effect might be and whether the authors have considered this effect in detail. For example, the asymmetry found in the Fraunhofer patterns could be a result of this. I am also wondering if the suppression of skewness reported in Fig. 2 could be a result of the random phases discussed above.

2. Phase shifts – The main claim of the paper is that a phase shifts are observed as a function of gate voltage with out changing field. While theory indeed predicts a phase shift due to spin-orbit interaction, I think it is important to properly rule out other possibilities before reaching this

conclusion. One specific concern is that the current flow through the JJ changes slightly with gate voltage which would then change the enclosed flux in the junction and loop. For example, a slight depletion or accumulation near the boundaries of the junction will have such an effect. Additionally, a change in the current distribution through the junction with density could also affect this phase. The authors plot all data as a function of inferred SQUID phase which makes it impossible to see if there are slight changes in the flux through the junction or SQUID. Finally I'd like to point out that the data in Fig. 1d looks identical to the data shown in arXiv:1906.01179 published by the same authors. While this is a characterization plot, it might be better to not repeat the same data in two publications.

Nature Communication: Answer to the referees

I. REFEREE 1: REMARKS TO THE AUTHORS

This paper reports on the fabrication of a SQUID based on two parallel hybrid Josephson junctions made from superconducting Aluminum electrodes deposited on an InAs quantum well. Two gates are used to control independently the carrier concentration and Rashba coefficient in the two Josephson junctions. Changing the gate voltage on one of the junction, they observe a phase-shift of the current-phase relationship when an in-plane magnetic field is present. This phase shift is attributed to the anomalous phase shift expected theoretically in presence of a finite Rashba coefficient and finite spinpolarization (here induced by the in-plane magnetic field). The data and their interpretation are convincing. The anomalous phase shift has been observed recently in two other systems, InSb nanowires [Ref 4] and Bi₂Se₃ [Ref 5]. However, as described in their introduction, this observation of the anomalous phase shift in 2D hybrid Josephson junction with InAs quantum wells is important and timely given the major role of the InAs platform for Majorana physics. Furthermore, this result on InAs, together with the recent result on Bi₂Se₃ [5], may motivate the search for the anomalous phase shift with many other materials presenting strong spin-orbit coupling. Because of this, I believe that this paper should be of interest to a large community and I recommend its publication in Nature Comm. I have some minor suggestions that the author may consider: Main text: Page 2, col 2, beginning 2 paragraph : Individual JJs are characterized in in-plane magnetic field (Add ref to Supplementary Fig. S1).

We thank the reviewer for their careful reading and we have added a reference to Fig. S1.

Main text: Page 2, col 2, beginning third paragraph : where $\Delta = ??$ is the superconducting gap of the Al. (Precise which numerical value you took for Δ)

We thank the reviewer for their careful reading. We have added the numerical value.

Method section A. Growth and fabrication: We deposit 50 nm of AlO₂ using atomic layer deposition to isolate gate electrodes. AlO₂ is not common, are you sure you didn't mean Al₂O₃ ? please check.

We thank the reviewer for their careful reading. We did mean Al₂O₃ and the text has been corrected.

II. REVIEWER #2 (REMARKS TO THE AUTHOR):

The paper "Gate Controlled Anomalous Phase Shift in Al/InAs Josephson Junctions" is an experimental paper reporting the observation of a tunable anomalous Josephson effect measured using a SQUID device. The anomalous Josephson effect refers to the possibility of having a finite Josephson current at zero phase difference between the superconducting leads, or the opposite, i.e. a finite phase difference at zero flowing current. There are very few experimental evidences of this phenomenon. Moreover, in addition with respect to previous papers, in the present manuscript the authors show a tunability of the effect using a voltage gate. I think that the paper deserves publication. However I have some points that I would like to be clarified prior to publication.

1) The authors compare with theoretical models explicitly addressing long junctions. It would be better to compare and cite the papers predicting the effect in the short junction limit: e.g. , J. Phys. Soc. Jpn. 82 (2013), Journal of Physics: Condensed Matter 27, 205301 (2015), Physical Review B 98, 144510 (2018).

We thank the reviewer for their suggestion and we agree that papers addressing the short limit are the most applicable to our results. To address this we have included reference to J. Phys. Soc. Jpn. 82 (2013), and Journal of Physics: Condensed Matter 27, 205301 (2015) along with additional text "Theoretical work in the short junction limit is generally restricted to nanowire systems with only a few conduction channels [J. Phys. Soc. Jpn. 82 (2013), Journal of Physics: Condensed Matter 27, 205301 (2015)]. This leads to a geometry that is drastically opposed to the current situation where $W \gg L$. However, we would emphasize that all work we are aware of that address the short limit either address nanowire geometries or include ferromagnetic materials. We believe the 2D nature of the proximitized region could play an important role in our devices. The inclusion of ferromagnetic materials [Physical Review B 98, 144510 (2018)] introduces magnetization orders of magnitude larger than in our system. As such quantitative comparisons are difficult.

2) In Fig. 2 the authors fit the I vs phi relation using Eq. 1, and conclude that the junction transparency is strongly affected by the external magnetic field. However they admit that the mechanism behind this dependence is not well understood. I am not sure that their approach is flawless here. Usually a correct approach to estimate the transmission of the Junction is performing excess current measurements (and using the very well known Blonder Tinkham Klapwijk theory). To me it looks quite unusual that a SNS system can have 0.1 transparency (that value sometimes can be found even in SIS junctions!). The point here is that the expression in Eq. 1 is well suited in the case of a single channel SNS junction in the ballistic limit. If I understand correctly, the junctions studied have a very large number of channels, each of them can have a different transparency, and extrapolating the junction transparency using eq. 1 can be a bit oversimplified.

We thank the reviewer for their comments. We have elaborated on our reasons for only using Eq. 1 to describe our junctions and why we think this approach captures all relevant physics as well as currently possible. We have added the text "It should be noted that this expression describes the transparency of a single ballistic channel. However in our devices there are many conduction channels present (~ 300) so the transparency we extract should be considered an average over all the channels. In the absence of disorder each channel can be considered individually. If we assume a Gaussian distribution of transparencies we approximately recover the single channel result for the mean transparency. In a more realistic system finite disorder can mix channels and substantially alter the junction properties as will be discussed in the context of an anomalous phase shift below." to the last paragraph on page 2.

3) I agree with the authors that they observe a large effect (large phi0). However their explanation in terms of the number of channels is only qualitative and not satisfactory. I suggest the authors to invest more efforts to give a microscopic description of their sample. They could be able to do that as there are theorists in the team!

Motivated by the referee's suggestion we took the time to study, using a microscopic model, the possible source of the large anomalous ϕ_0 observed experimentally in SQUID devices. Below, we present the results of our numerical studies, and compare the results with known analytic formulas.

First we would like to underline that the existing literature discussing the anomalous phase in Josephson junction is limited to systems in which the induced superconducting gap is small, either because the temperature is considered to be close to T_c or because the transparency of the S-N interface is low (Ref 26, 28) or to quasi-1D systems (Ref 25, 27, 29, 30). As a consequence none of their results are directly applicable to our case of a 2D junction with a strong induced gap.

Using tight-binding simulations based on Kwant [Groth, C. W., Wimmer, M., Akhmerov, A. R., Waintal, X., Kwant: a software package for quantum transport, New J. Phys. **16**, 063065, 2014], we have attempted to extend those results to 2D junction with transparent interfaces. The junction was modeled as a finite size system of width W with two superconducting sections of length L_{SC} placed on the side of a normal region of length L . To avoid artificial finite-size effects we used $L_{SC} \gtrsim 10\xi$, where ξ is the superconducting coherence length of the superconductor.

In the regions of the 2DEG in contact with epitaxial Al layer, that is to say the regions that are strongly proximitized, we assumed that the chemical potential was pinned and neglected the Zeeman term and the spin-orbit coupling (SOC). The proximity induced gap was introduced as a pairing potential term in the tight-binding model.

We computed the spectrum of the finite size system as a function of the imposed phase difference ϕ between the two superconducting sections. Assuming $T \rightarrow 0$, knowledge of the spectrum $\{\epsilon_n(\phi)\}$ allows us to calculate the Josephson current as:

$$J = -\frac{2e}{\hbar} \sum_{\epsilon_n < 0} \frac{\partial \epsilon_n(\phi)}{\partial \phi} \quad (1)$$

It should be noted that a full microscopic model of the experimental setup is very challenging particularly when considering many transverse modes. It is computationally very demanding, and beyond our current computational resources, to consider systems with dimensions comparable to the experimental ones. This is in part due to the fact that an accurate estimate of ϕ_0 requires taking into account the contributions of the states in the continuum, outside the superconducting gap. This requires knowledge of almost the full spectrum of the Hamiltonian and therefore strongly reduces the advantage of the computational speed-up provided by sparse-matrix diagonalization methods.

Comparison of the values obtained from simulations with the experiment reveal that, in the ballistic regime, the calculated value of ϕ_0 is orders of magnitude smaller than the value measured experimentally, even when all the transverse modes are taken into account.

As an example, figures 1 (a), (b), show the spectrum and Josephson current, respectively, as function of ϕ , for the long ballistic case. We observe a small anomalous phase shift $\phi_0 \approx 0.02$ using parameters in the range of experimental numbers.

FIG. 1: (a) shows the spectrum as a function of ϕ of a Josephson junction in the ballistic regime with $L = 100$ nm, $W = 200$ nm, $\xi = 30$ nm, $E_z = (1/2)\Delta$, and $\alpha = 0.15$ eVÅ, $m^* = 0.04m_e$ for $n = 8.3 \times 10^{11}$ cm $^{-2}$. (b) Shows the corresponding Josephson current $J(\phi)$. The inset shows the $J(\phi)$ close to $\phi = 0$.

Even though the existing literature does not apply to our experimental situation, it is clear from it that junctions in the diffusive regime are always expected to display a larger phase-shift than junction in the ballistic regime. We are therefore led to conclude that this anomalous phase-shift must be due to the modes for which the transport is diffusive or pseudo-diffusive.

For the type of devices used in the current experiment, the mean free path l_{mfp} is about 100 – 200 nm, see Ref 20 and 21 of the main text. This implies that most of the modes in the normal region of the Josephson junction are in an intermediate regime, making diffusive transport in some subset of the channels quite plausible. However the numerical study of the diffusive regime would require to perform a disorder average and is beyond our current computational means. A more in-depth theoretical study could be the focus of a future work.

Minor points:

1) The references are not sorted.

We thank the reviewer for noticing this issue. The references are now properly sorted.

2) Caption of Fig. 2 The authors refer to panel (c). I guess it should be panel (b)

We thank the reviewer for their careful reading and have corrected the caption for Fig. 2.

III. REVIEWER 3 (REMARKS TO THE AUTHOR):

The work by Mayer et al, reports measurements of Josephson junctions using InAs QW's coupled to epitaxial Aluminum. The work looks to establish what has been predicted theoretically in the literature that a phase shift in the current phase relation should develop due to spin-orbit interaction. The work uses a SQUID geometry to determine the phase. The observed phase shift appears to be considerably larger than what theory predicts in the limit of junction length larger than the coherence length. The authors speculate that their work is in the opposite limit which is outside the range of validity of existing theory. While the experiments are carefully performed, I feel there is more evidence needed in order to ensure the observed phase shifts are indeed a result of spin-orbit interaction and not some other less interesting effect. Below I provide some of my concerns and comments:

1. Effects of parallel field on Fraunhofer patterns - In page 1 the authors enter a discussion on the behavior of the system as a function of B_x and write: “Measurements of Fraunhofer pattern with increasing in-plane field show increasing asymmetry, but unlike previous studies this asymmetry is found to be independent of in-plane field direction. Additionally, despite the distortions, the Fraunhofer pattern appears to remain periodic, indicating a homogeneous current distribution at all fields. Figures and further discussion are presented in Supplemental.” I could not follow this statement and was unable to find the discussion in the supplemental material on current distribution.

We thank the reviewer for their comments and to clarify the matter we have rewritten the quoted section to read “Unlike previous studies this asymmetry is found to be independent of in-plane magnetic field direction.

Additionally, despite these distortions, the Fraunhofer pattern appears to remain periodic. Significant changes in current distribution, such as edge conduction, should alter the periodicity, specifically with respect to the central Fraunhofer peak. The absence of any such effect indicates a homogeneous current distribution at all fields.” to clarify our message. We have also expanded the ”Fraunhofer” section in the supplemental text. We have added ”Despite these distortions a clear central peak remains at all magnetic fields below B_c . Additionally, as stated in the main text, the period of Fraunhofer oscillations is unchanged. This indicates there are not large deviations from a uniform current distribution even in the presence of large in-plane magnetic fields.” to emphasize our the importance of these results for current distribution.

The effects of B_x are complex due to what is known as the Doppler effect. The fact the SC is above the 2DEG leads to a unique winding of the phase along the junction, when an in-plane field is present and applied perpendicular to the edge of the SC. This could modify the Fraunhofer pattern substantially. While this Doppler effect should be strongest for B applied along X (perpendicular to the edge of the SC) it could also have an increasing effect when the field is applied along Y . This comes in as random phases generated along the junction due to lithographic roughness of the superconducting edges. The effect along X is known to strongly modify the current flow pattern in particular near the edges of the junction. I am therefore wondering what some of the consequences of this effect might be and whether the authors have considered this effect in detail. For example, the asymmetry found in the Fraunhofer patterns could be a result of this. I am also wondering if the suppression of skewness reported in Fig. 2 could be a result of the random phases discussed above.

We thank the reviewer for bringing this effect to our attention. We have studied the effect of in-plane magnetic field in both directions extensively. In this structure it is expected that the InAs region directly below the Al is strongly proximitized and has a negligible spin-orbit coupling and Zeeman energy at all fields. This means that the bound states that exist in the weak link should have negligible overlap with the Al, minimizing coupling with the supercurrent that is responsible for the Doppler effect. Our results indicate that the area’s as measured through the period of Fraunhofer and SQUID oscillations remains the same up to a few percent at all fields.

We should also note that an imperfect interface of the S-N interface could give rise to “local Doppler shifts”, i.e. local shifts of the quasiparticle energies and random phases. In our setup, as mentioned earlier, all the sections of the JJ, S sections, and N section, are in the same 2D InAs layer: the S sections are the regions of InAs layer proximitized by the Al. Because of this setup we expect the S-N interfaces to be much cleaner than in setups in which the SC and the normal regions are two different materials: the irregularities in the termination of the Al layer, due to screening, result in much less irregular S-N interfaces in the InAs layer. As a consequence we expect the local shifts due to the Doppler effect to be quite small.

2. Phase shifts – The main claim of the paper is that a phase shifts are observed as a function of gate voltage with out changing field. While theory indeed predicts a phase shift due to spin-orbit interaction, I think it is important to properly rule out other possibilities before reaching this conclusion. One specific concern is that the current flow through the JJ changes slightly with gate voltage which would then change the enclosed flux in the junction and loop. For example, a slight depletion or accumulation near the boundaries of the junction will have such an effect. Additionally, a change in the current distribution through the junction with density could also affect this phase. The authors plot all data as a function of inferred SQUID phase which makes it impossible to see if there are slight changes in the flux through the junction or SQUID.

We agree that it is important to rule out other possibilities. We had plotted all data against inferred phase for the readers convenience but to alleviate any concerns we have added plots Fig. S2 to the supplemental which show SQUID oscillations as a function of magnetic field. We would also like to point out that as indicated in the text all fitting of the experimental data is accomplished using a single frequency for the SQUID oscillations, and hence the phase we indicate is related to the field applied by a simple linear relation.

Furthermore, we can estimate the relative surface change that the gate would need to induce to generate the observed phase-shift. The derivation has been added to the Supplementary Materials (section III) and we simply point out here that to obtain a phase-shift of $\pi/2$, the gate would have to induce a relative surface change of the order of 5%. This is not realistic since the SQUID area is $25 \mu m^2$ and each junction is about 100 nm by $1 \mu m$. The impact of flux focusing on the junction can be estimated to give only twice the field expected in the junction based on the Fraunhofer periodicity which allows us to rule out this possibility.

Finally I’s like to point out that the data in Fig. 1d looks identical to the data shown in arXiv:1906.01179 published by the same authors. While this is a characterization plot, it might be better to not repeat the same data

in two publications.

We thank the reviewer for this suggestion. We have removed Fig. 1d from the manuscript.

REVIEWERS' COMMENTS:

Reviewer #1 (Remarks to the Author):

I believe that the authors provided reasonable responses to my comments as well as to the comments from other referees. The paper can be published as it is.

Reviewer #2 (Remarks to the Author):

The authors has answered satisfactorily to all my points.
I confirm my previous report and suggest to publish the manuscript.

Nature Communication: Answer to the referees

I. SECOND ROUND OF REVIEW

Reviewer 1 (Remarks to the Author):

I believe that the authors provided reasonable responses to my comments as well as to the comments from other referees. The paper can be published as it is.

Reviewer 2 (Remarks to the Author):

The authors has answered satisfactorily to all my points. I confirm my previous report and suggest to publish the manuscript.

We like to thank the referees for their encouraging comments and recommending our paper for publication.